# Clinical and Neural Predictors of Treatment Response to Music Listening Intervention after Stroke

**DOI:** 10.3390/brainsci11121576

**Published:** 2021-11-29

**Authors:** Aleksi J. Sihvonen, Teppo Särkämö

**Affiliations:** 1Cognitive Brain Research Unit, Department of Psychology and Logopedics, Faculty of Medicine, University of Helsinki, 00014 Helsinki, Finland; teppo.sarkamo@helsinki.fi; 2Queensland Aphasia Research Centre and UQ Centre for Clinical Research, School of Health and Rehabilitation Sciences, The University of Queensland, Brisbane, QLD 4029, Australia

**Keywords:** stroke, language, music, intervention, treatment response, treatment predictor

## Abstract

Patients with post-stroke impairments present often significant variation in response to therapeutic interventions. Recent studies have shown that daily music listening can aid post-stroke recovery of language and memory, but reliable predictors of treatment response are unknown. Utilizing data from the music intervention arms of a single-blind randomized controlled trial (RCT) on stroke patients (N = 31), we built regression models to predict the treatment response of a two-month music listening intervention on language skills and verbal memory with baseline demographic, clinical and musical data as well as fMRI data from a music listening task. Clinically, greater improvement in verbal memory and language skills after the music listening intervention were predicted by the severity of the initial deficit and educational level. Neurally, greater baseline fMRI activation during vocal music listening in the left parietal cortical and medial frontal areas predicted greater treatment-induced improvement in language skills and greater baseline engagement of the auditory network during instrumental music listening predicted improvement in both verbal memory and language skills. Our results suggest that clinical, demographic, and neuroimaging data predicts music listening treatment response. This data could be used clinically to target music-based treatments.

## 1. Introduction

Music-based interventions have received growing interest in rehabilitation of neurological diseases, especially stroke, during the last two decades [1]. This has been driven on the one hand by the rapidly increasing prevalence of stroke as well as its massive socioeconomic burden and growing need for health-economically ideal rehabilitation tools [2] and on the other hand by the advances made in the field of music neuroscience in uncovering the brain mechanisms underlying our ability to perceive, experience, and produce music [3,4].

Rehabilitation aims to achieve functional restoration, which neurally relies upon the ability of spared neurons to compensate for the lost function by rebuilding and remodeling the injured networks [5,6]. These recovery processes are activity-dependent, meaning that they require stimulation of the neural networks subserving the rehabilitated function to bring about beneficial behavioural change. In this context, music can be viewed as a form of environmental enrichment that increases activity-dependent neuroplasticity in the large-scale brain network it stimulates [7,8,9], thus promoting synaptic plasticity and increasing neurotrophic factor levels, supporting the recovering brain [10,11,12].

Previously, music has been found to be a useful tool to aid post-stroke recovery. During the last 20 years, promising evidence has emerged on the efficacy of music-based interventions targeted for specific cognitive, motor, and verbal deficits caused by stroke [1]. These include active interventions that utilize instrument playing or musical rhythm to rehabilitate deficits in motor control of movements (hemiparesis) [13] and singing-based methods to rehabilitate deficits in speech production (aphasia) [14] as well as receptive interventions that utilize music listening. In three randomized controlled trials (RCTs), daily music listening during the first three post-stroke months has been reported to enhance the recovery of verbal memory, attention, and language skills compared to a control intervention (audiobook listening) or standard care [15,16,17]. Using structural and functional magnetic resonance imaging (s/fMRI), these positive behavioural effects have been coupled with structural neuroplasticity in prefrontal, temporal, and limbic brain regions and white matter tracts [17,18,19], as well as with increased functional activation or connectivity in motor cortical, temporal, and parietal regions [17,19].

Despite this evidence, little is currently known about the neural and behavioural prerequisites of successful music listening based treatment response. There is growing interest in predicting post-stroke functional outcomes using demographic, clinical, and neurological variables [20] as well as neuroimaging data [21]. In general, treatment response evaluation combining clinical and neuroimaging information could be used in the clinical work to improve care and personalize rehabilitation plans. Crucially, this could help to better target rehabilitation strategies, reduce unwanted treatment response variation, and increase equity of access to limited rehabilitation services. Identifying these features in music-based intervention methods could help their translation into clinical practice as well as improve patient outcomes via easily applicable and cost-effective rehabilitation strategies that could complement traditional methods such as physiotherapy, occupational therapy, and speech therapy.

The present study sought to identify which clinical, demographic, and musical factors and music-evoked functional activation patterns and networks were the best treatment response predictors of a post-stroke music listening intervention. To do so, we assessed a subsample of 31 stroke patients from our latest RCT who received a music listening intervention [17,19] and used regression models with subacute (baseline) post-stroke stage clinical, demographic, and musical information as well as music listening induced task-based functional MRI (fMRI) activity and network-level functional connectivity to predict favorable language and verbal memory outcomes after the music intervention. At the brain level, we hypothesized that greater neural responsiveness to music at the early post-stroke stage, indicated by stronger activity or connectivity in language- and auditory-related regions in the music task, would be associated with favorable treatment outcomes.

## 2. Materials & Methods

### 2.1. Subjects and Study Design

Fifty patients with an acute ischaemic stroke or intracerebral hemorrhage in the left or right hemisphere were recruited in 2013–2016 at the Division of Clinical Neurosciences of Turku University Hospital to a three-arm single-blind randomized controlled trial (RCT). Inclusion criteria were acute unilateral stroke, right-handedness, <80 years of age, capability to communicate in Finnish, residence in Southwest Finland, ability to co-operate, and normal hearing. Exclusion criteria were prior neurological or psychiatric disease and substance abuse. All patients gave an informed consent, and the study was approved by the Ethics Committee of the Hospital District of Southwest Finland and performed in conformance with the Declaration of Helsinki. All patients received standard stroke treatment and rehabilitation. Baseline (subacute stage) MRI scans and behavioural assessments were performed <3 weeks post-stroke (mean 12 days), followed by randomization to vocal music (VMG, N = 17), instrumental music (IMG, N = 17), and audio book groups (N = 16). Post-intervention follow-up behavioural assessments and MRI scans were performed at the three-month post-stroke stage. All assessments were performed blinded to the group allocation of the patients.

For the purposes of this study, we focused on patients who received music listening treatment, that is, patients in the VMG and IMG arms of the trial. At 3-month post-stroke stage, 31 patients (VMG, N = 14; IMG, N = 17) had completed MRI imaging as well as behavioural assessments at both the subacute and 3-month follow-up stage and were included in the present analyses for predicting music listening treatment responses (Table 1).

### 2.2. Intervention

The intervention protocol has previously been described in detail [15,17]. In short, the patients were contacted by a professional music therapist after baseline assessments, who informed them of the group allocation. The listening material in all group allocations was selected individually to match the music preferences of the patient as closely as possible, that is, the music therapists provided the patients in the music groups with portable player, headphones, and the patient’s own favorite vocal (in VMG) or instrumental (in IMG) music in any musical genre. The patients were trained to use the players and instructed to listen to the material by themselves daily (min. 1 h per day) for the following two months and to keep a listening diary. During the intervention period, the music therapists kept close weekly contact with the patients to encourage listening, to provide more material, and to give practical aid in using the equipment, if needed. After the intervention period, the patients were free to continue listening at their own will.

### 2.3. MRI Data Acquisition and Preprocessing

All patients were scanned on a 3T Siemens Magnetom Verio scanner with a standard 12-channel head matrix coil at the Department of Radiology of Turku University Hospital. High-resolution T1-weighted anatomical images and a task-fMRI scan using a single-shot T2*-weighted gradient-echo EPI sequence (280 functional volumes; 32 slices; slice thickness = 3.5 mm; TR = 2010 ms; TE = 30 ms; flip angle = 80°; voxel size = 2.8 × 2.8 × 3.5 mm^3^) were acquired at both time points. During the task-fMRI scan (a block design), the patients were presented with 15-s excerpts of well-known Finnish songs with (i) sung lyrics (Vocal, 6 blocks) and (ii) without sung lyrics (Instrumental, 6 blocks), (iii) well-known Finnish poems (Speech, 6 blocks), and (iv) no auditory stimuli (Rest, 18 blocks) through MR-compatible headphones using Presentation software (Neurobehavioural Systems, version 16.3). The rest blocks were presented in between the auditory blocks. To avoid stimulus order predictability, the order of the auditory blocks was randomized across subjects and time, and intervention listening material was not used in the task-fMRI excerpts.

MRI data were preprocessed using Statistical Parametric Mapping software (SPM8, Wellcome Centre for Human Neuroimaging, University College, London, UK, https://www.fil.ion.ucl.ac.uk/spm/ accessed on 13 January 2020) under MATLAB version 8.4.0. The fMRI images were first realigned, and a mean image all functional volumes were created. This mean image was then coregistered to the T1-weighted image which were reoriented according to the anterior commissure. Cost function masking was applied to achieve optimal normalization of the stroke-lesioned brain tissue, with no out-of-brain distortion or post-registration lesion shrinkage [22,23,24]. Cost function masking was performed by manually depicting the stroke lesions slice-by-slice to the individual T1-weighted images using MRIcron (http://people.cas.sc.edu/rorden/mricron/index.html accessed on 13 January 2020) [25]. All lesion tracing was carried out by one person (author A.J.S.) experienced in this matter [26,27]. Task-fMRI data were normalized to Montreal Neurological Institution (MNI) space using Unified Segmentation [28] and re-sampled into isotropic 2 × 2 × 2 mm^3^ voxel size. Lastly, the preprocessed fMRI data were smoothed using an isotropic spatial filter (FWHM = 8 mm).

The statistical evaluation of the task-fMRI data was based on a least-square estimation using the general linear model in both time points (Subacute/3-month). At the individual level, the different task conditions (Vocal/Instrumental/Speech) were modelled with a box-car regressor waveform convolved with a canonical hemodynamic response function. Data were high-pass filtered to a maximum of 1/128 Hz and serial autocorrelations were estimated using an autoregressive model [AR(1) model]. In addition, confounding factors from head movement were included in the model. A block-related design matrix was created including the conditions of interest (Vocal/Instrumental/Speech). After model estimation, main effects for each condition against rest were calculated (e.g., Vocal > Rest).

### 2.4. Network-Level Functional Connectivity

Group-level spatial independent component analysis (ICA) was performed using the Group ICA of fMRI Toolbox (GIFT) software (http://mialab.mrn.org/software/gift/ accessed on 13 January 2020). The ICA spatial components present during the task-fMRI data acquisition were extracted using the whole run. First, the intensity of the acquired images was normalized. Second, the data were concatenated and reduced to the 20 temporal dimensions representing independent components (networks) using principal component analysis [9,17,29,30,31]. The data were analyzed using the infomax algorithm [32] and no scaling was applied since the intensities of the acquired spatial maps are in percentages of signal change. The acquired components were inspected visually to detect deficits and artefacts (e.g., noise). From the ICA spatial components representing the different networks, the auditory network, left and right frontoparietal network, language network and default mode network were identified and selected for further analyses. All these networks have been previously associated with music perception [9] or favorable treatment outcomes after music-based interventions [17,19]. The time course of the selected networks was fitted to an SPM model that included the Vocal, Instrumental, and Speech conditions as regressors, yielding beta values representing the engagement of each network during each condition.

### 2.5. Behavioural Outcome Measures

As music listening has been shown to primarily improve verbal memory and language recovery after stroke [15,16,17], we focused here on predicting the favorable outcome in these two domains. Following our previous studies [15,17], a language skills composite score was derived from a summary score calculated by adding up the raw test scores of the standard Verbal Fluency Test [33], the shortened Token Test [34], and the shortened Boston Naming test (a 20-item version including every third of the original 60-line drawings with a maximum score of 20) [35,36]. Verbal memory composite score derived from raw test scores of word-list learning task (a list of 10 words, immediate and delayed recall) and the Rivermead Behavioural Memory Test story recall (immediate and delayed recall) tasks [37]. At the subacute (baseline) stage, also the National Institutes of Health Stroke Scale (NIH-SS) [38], the Aphasia Severity Rating Scale of the Boston Diagnostic Aphasia Examination (BDAE-ASRS) [39], the Scale and Rhythm subtests of the Montreal Battery of Evaluation of Amusia (MBEA) [40], and the Barcelona Music Reward Questionnaire (BMRQ) [41] were administered.

### 2.6. Statistical Analysis

First, we evaluated which clinically relevant baseline information best predicted the post-intervention (three-month) treatment response by calculating two stepwise regression models in IBM SPSS Statistics 27 separately for the longitudinal change (three months—subacute stage) in language skills and verbal memory (dependent variables). As independent (regressor) variables, we included three types of variables: demographic, clinical, and musical. The demographic variables were age, gender, and years of education. The clinical variables were stroke type (infarct/hemorrhage), lesion size, lesion laterality, stroke severity (NIH-SS total score), and aphasia severity (BDAE-ASRS score). The musical variables were music perception ability (MBEA Scale and Rhythm subtest average score) and self-reported reward value of music in daily life (BMRQ) total score). In addition, the baseline language skills/verbal memory score, respectively, was added in the model as a regressor.

Next, we evaluated if the brain’s initial responsiveness to music as indicated by baseline fMRI activity in a music listening task could predict treatment response. The statistical analyses of the preprocessed task-fMRI data were carried out using SPM8. Four regression models were calculated: longitudinal change (three months—subacute stage) in (i) language skills score and (ii) verbal memory score were used as dependent variables and focal BOLD signal levels during (1) vocal and (2) instrumental conditions as regressors. Statistical maps were thresholded in SPM at a voxel-level uncorrected *p* < 0.005 threshold and standard SPM FWE cluster-level correction based on RFT with a *p* < 0.05 was used [17]. Subacute stage language/verbal memory score, total intracranial volume and lesion size were used as nuisance covariates. Neuroanatomical areas were identified using the Automated Anatomical Labeling Atlas provided within the xjView toolbox (http://www.alivelearn.net/xjview/ accessed on 1 November 2020).

Lastly, we evaluated whether subacute stage functional network engagement could predict music listening treatment response. The statistical analyses of the network-level functional connectivity were analyzed by extracting the beta values for each patient per each network and then imported then into SPSS. Following the task-fMRI analyses, two stepwise regression models were calculated: longitudinal change (three months—subacute stage) in (1) language skills score and (2) verbal memory score were used as dependent variables and mean network engagement in the auditory network, left and right frontoparietal network, language network and default mode network during Vocal and Instrumental condition as regressors. In addition, subacute stage language skills and verbal memory score, respectively, was added to the model as a nuisance covariate.

## 3. Results

First, we evaluated which clinical, demographic, and musical factors would predict the favorable music listening treatment response. In the stepwise regression model, the improvement in verbal memory was predicted by the baseline (subacute) verbal memory score and years of education (Table 2). The improvement in language skills was solely predicted (negatively) by subacute aphasia severity (that is, more severely aphasic patients improved more in language skills). There were no other significant predictors.

For the focal fMRI activations, greater subacute stage activity in the vocal condition in the left parietal areas (inferior parietal lobule, supramarginal gyrus and postcentral gyrus) as well as bilaterally in medial frontal areas (supplementary motor area and cingulate gyrus) predicted greater longitudinal improvement in language skills (Table 3, Figure 1A). No significant predictors were observed for improvement in verbal memory.

For the functional networks, greater subacute stage engagement of the auditory network in the Instrumental condition predicted greater longitudinal improvement in both language skills and verbal memory (Table 4, Figure 1B). Additionally, the subacute language skills/verbal memory score also emerged as a significant predictor in the model.

## 4. Discussion

This study set out to determine the subacute clinical and functional neuroimaging predictors of favorable language and verbal memory outcome after a post-stroke music listening intervention. Our two main findings were that (i) greater activity in the left inferior parietal cortex and bilateral medial prefrontal regions during subacute stage vocal music listening predicted the treatment response for language skills; and (ii) greater engagement of the auditory network during subacute stage instrumental music listening predicted the treatment responses for language skills and verbal memory. In addition, the initial severity of both language and verbal memory deficits predicted the treatment responses. The present study extends previous results on the rehabilitative effects of music listening after stroke [15,16,17,18,19] by revealing important information of possible treatment responders. This putative evidence is important in personalizing music-based rehabilitation strategies [1] as well as in improving our understanding of post-stroke language and verbal memory rehabilitation.

Interest towards prediction tools for stroke rehabilitation has dramatically increased during the last few years [20]. Recovery from post-stroke language and cognitive deficits depends on a number of factors, including, for example, patient demographics, initial stroke severity, infarct/hemorrhage characteristics, and pre-stroke brain status. Factors modifying the recovery process, such as the rehabilitation intensity, frequency, and timing, have also a strong impact on the outcome. However, current post-stroke language treatments have shown highly variable outcomes and the evidence for their long-term effects is scarce [42]. Moreover, patients often receive treatment, which is insufficient in quantity and predominantly given outside the optimal time window for brain plasticity. Despite this, only little research has focused on evaluating treatment response predictors after post-stroke language deficits. An accurate prediction of neurocognitive recovery after stroke is crucial to provide rehabilitation strategies based on the patients’ individual needs. Moreover, as most therapeutic interventions are most beneficial when started early after stroke, research should provide information regarding the early-stage predicators to provide the patients with most beneficial therapeutic interventions [43]. In this vein, evaluating post-stroke music listening intervention treatment response predictors is crucial, as it provides a complementary rehabilitation strategy that could be safely implemented in the early stages of stroke rehabilitation and seems to specifically target the recovering language network [19] and functions [15,17].

The present results suggest that treatment response in language skills/verbal memory after music listening intervention was predicted behaviourally by demographic (years of education) and clinical (initial severity of the language/verbal memory deficit) data. In other words, patients with a more severe initial deficit and more education (for verbal memory only) benefited more from the music listening intervention. This seems to be at odds with findings from previous studies in aphasia where more severe aphasia has been linked to worse outcome in spontaneous recovery and in response to speech and language therapy [44]. Our result could be interpreted as indicating that music listening might be particularly beneficial and useful in patients with higher severity of aphasia, but then again it may also reflect the current study population, which did not include patients with global or very severe aphasia. Moreover, as the current sample size is limited, future studies with larger samples of patients representing the full continuum of aphasia severity are needed to address this issue. In the verbal memory domain, but not in language skills, the treatment response was also associated with years of education. This is in line with previous research showing that higher education level is linked to better recovery of memory post-stroke, supporting the general notion that education may increase the capacity/efficiency of cognitive networks and result in greater “reserve” after injury [45]. Interestingly, music perception ability (MBEA) and subjective reward value of music (BMRQ) were not associated with treatment outcomes, suggesting (optimistically) that music listening is therapeutically beneficial regardless of the musical skills or background of the patients. This may be specific to a receptive music intervention as BMRQ scores have been reported to correlate with the motor efficacy of an active music intervention involving instrumental training (music-supported therapy, MST) [46].

The fMRI results suggest that the music listening treatment response can be predicted by both task-based fMRI as well as functional network analyses. Patients showing a favorable response in language skills to the music listening intervention elicited increased BOLD activity in the left parietal (inferior parietal lobule, supramarginal and postcentral gyrus) as well as bilateral medial prefrontal areas (supplementary motor area and cingulate gyrus) during subacute stage vocal music listening. The left inferior parietal lobule and the supramarginal gyrus have been shown to integrate information during language processing within three functional networks: peri-sylvian, fronto-parietal and default-mode network [47]. Together, these networks form the canonical semantic network that supports language and memory functions in post-stroke aphasia [48]. The present results suggest that preserved ability to engage the critical inferior parietal hubs for this network during vocal music listening predict favorable post-stroke music listening treatment response. Moreover, favorable outcomes were associated with higher activity in the medial prefrontal areas that are engaged by vocal music listening and have been implicated in the auditory-motor processing of music [7] and singing [49]. After stroke, higher activation in the left supplementary motor and inferior frontal areas during a language task have been shown to correlate with language performance [50]. Structurally, these areas are strongly interconnected via the left frontal aslant tract which is an important tract for speech production [51,52,53]. We have previously shown that daily post-stroke vocal music listening increases left frontal aslant tract fractional anisotropy values which were linked to better recovery of language skills [19].

Favorable treatment responses in language skills and verbal memory were also predicted by greater engagement of the auditory network during subacute stage instrumental music listening. Previous evidence has shown that measuring preserved fMRI activity in the primary auditory areas is a robust and reliable method to predict post-stroke language improvement, even in aphasic patients [43]. While predicting language improvement using fMRI data has shown the highest accuracy when the analyses are focused on the key language areas [43], these areas are usually affected by the initial stroke lesion and diaschisis whereas the primary auditory areas are less likely to be affected by functional disconnections. Therefore, in the early stages of stroke, the auditory network engagement might provide outcome-relevant information compared to the language-specific effect. The present results support this by showing that the auditory network engagement during instrumental, but not during vocal music listening, predicts treatment responses. This may reflect a general preservation of the auditory system to respond to music whereas the task-fMRI results provide more specific information regarding responsiveness to linguistic and motor elements of vocal music in left parietal and bilateral frontal regions.

To our best knowledge, treatment response predictors of music-based interventions after a brain injury (e.g., stroke) have not been published nor studies that evaluate neurobiological treatment response predictors. However, the present results might be extended to other forms of music-based interventions such as music therapy where active and receptive music-based interventions are combined and delivered in a clinical setting by a credentialed music therapist. For example, in motor rehabilitation, the initial severity of paresis and activity in the motor and somatosensory cortices during music listening might predict treatment outcome in music therapy or MST. Similarly, in singing-based speech training interventions, such as melodic intonation therapy (MIT), preserved neural networks subserving language and singing could predict treatment responses. In music therapy, the therapeutic relationship might also play a significant role. For example, in autism spectrum disorder, the therapeutic relationship has been shown to be an important predictor of favourable outcome after music therapy [54]. In contrast, no such influence was identified in individual music therapy for depression, nor did age or musicianship predict significant responses [55]. Evaluating clinical and neurobiological predictors of treatment response to music therapy after stroke would be of great importance for further improving our understanding of stroke rehabilitation with music-based interventions.In conclusion, our findings suggest that the behavioural gains on language and memory induced by a daily music listening intervention after stroke [13,14,15] are associated with greater initial severity of the language/memory deficit, higher education level, and greater baseline music-evoked connectivity in the auditory network and activation in frontoparietal areas. Clinically, this may indicate that especially stroke patients with more severe language and memory deficits, higher cognitive reserve capacity, and greater neural responsiveness to music stimuli could benefit most from daily music listening during the first post-stroke months. However, as a limiting factor of the present study, it should be borne in mind that the results are based on a relatively small patient sample and the findings are correlational not causal. The broader predictive value and validity of the results remains to be proven with future larger-scale studies that can also more reliably predict outcomes on a single-subject level in a clinical setting.

## Figures and Tables

**Figure 1 brainsci-11-01576-f001:**
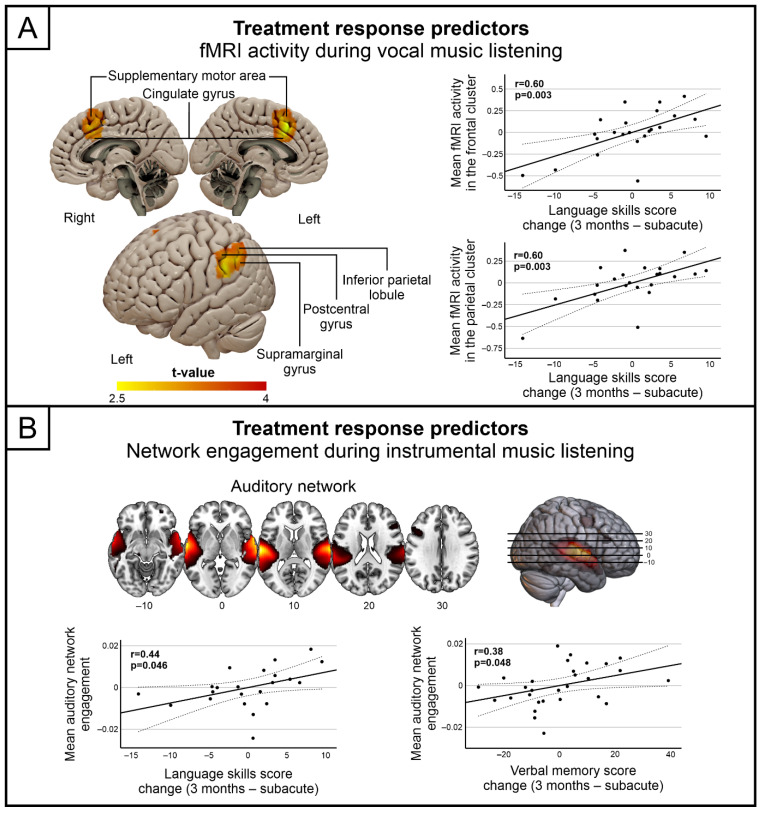
Significant music listening treatment response predictors. (**A**) fMRI activations during vocal music listening and (**B**) auditory network engagement during instrumental music listening at subacute stage predicting improved language and verbal memory outcomes after music listening treatment (3 months > subacute). Correlations to change in language skills/verbal memory are shown with scatter plots.

**Table 1 brainsci-11-01576-t001:** Baseline demographic, clinical, and musical characteristics of the patients (N = 31).

Demographic information	
Age (years)	55.4 (13.4)
Sex (female/male)	13/18
Education (years)	14.3 (3.7)
Clinical information	
Stroke type (infarct/hemorrhage)	22/9
NIH Stroke Scale score	6.0 (4.0)
Lesion size (dL)	0.6 (0.5)
Lesion laterality (left/right)	15/16
BDAE-ASRS score	4.5 (0.7)
Musical information	
MBEA Scale & Rhythm subtests average score	71.7 (15.8)
BMRQ score	73.5 (12.1)
Baseline behavioural outcome scores	
Language skill score (composite)	55.8 (8.7)
Verbal memory score (composite)	48.5 (18.7)

Data are mean (SD) unless otherwise stated. BDAE-ASRS = Boston Diagnostic Aphasia Examination/Aphasia Severity Rating Scale; BMRQ = Barcelona Music Reward Questionnaire; MBEA = Montreal Battery of Evaluation of Amusia; NIH = National Institutes of Health.

**Table 2 brainsci-11-01576-t002:** Clinical and demographic predictors of music listening treatment response.

Model	Variable	Beta	T	*p*	F(df)	R^2^	R^2^ Change
Verbal memory improvement							
1	Verbal memory score subacute	−0.513	−3.160	0.004	F(1,28) = 9.984	0.263	0.263
2	Verbal memory score subacute	−0.498	−3.333	0.001	F(2,27) = 8.928	0.398	0.135
	Education years	0.368	2.463				
Language skills improvement							
1	BDAE score subacute	−0.622	−3.554	0.002	F(1,20) = 12.630	0.387	0.387

Statistical information presented: Beta = standardized regression coefficient; T = t value; F(df) = F value (degrees of freedom); *p* = *p*-value; R^2^ = R Square; R^2^ change = R Square change.

**Table 3 brainsci-11-01576-t003:** Significant task-fMRI predictors of music listening treatment response.

Condition	Outcome	Area Name	Coordinates	Cluster Size	t-Value	Correlation
Vocal music listening	Language skills improvement	Left Supramarginal Gyrus (BA 40)	−60 −28 43	617	5.47	r = 0.60, *p* = 0.003
		Left Inferior Parietal Lobule (BA 40)	−54 −36 37			
		Left Postcentral Gyrus (BA 1, 2, 3)	−60 −21 29			
		Left Supplementary Motor Area (BA 6, 8)	−2 18 54	645	4.76	r = 0.60, *p* = 0.003
		Left Cingulate Gyrus (BA 24, 32)	−13 10 37			
		Right Supplementary Motor Area (BA 6, 8)	3 18 59			
		Right Cingulate Gyrus (BA 24, 32)	3 18 37			

All results are thresholded at voxel-wise *p* < 0.005 and cluster-wise FWE *p* < 0.05.

**Table 4 brainsci-11-01576-t004:** Functional network predictors of music listening treatment response.

Model	Variable	Beta	T	*p*	F(df)	R^2^	R^2^ Change
Verbal memory improvement							
1	Verbal memory score subacute	−0.513	−3.102	0.004	F(1,27) = 9.622	0.263	0.263
2	Verbal memory score subacute	−0.515	−3.301	0.003	F(2,26) = 7.564	0.368	0.105
	Auditory network engagement during instrumental music listening	0.324	2.079				
Language skills improvement							
1	Language skills score subacute	−0.537	−2.848	0.010	F(1,20) = 8.114	0.289	0.289
2	Language skills score subacute	−0.559	−3.209	0.005	F(2,19) = 7.053	0.426	0.137
	Auditory network engagement during instrumental music listening	0.371	2.133				

Statistical information presented: Beta = standardized regression coefficient; T = t value; F(df) = F value (degrees of freedom); *p* = *p*-value; R^2^ = R Square; R^2^ change = R Square change.

## Data Availability

The conditions of our ethics approval do not permit public archiving of anonymized study data. Readers seeking access to the data should contact the lead author A.J.S. or the local ethics committee at the University of Turku, Finland.

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
