# Peer review of "Clinical and Neural Predictors of Treatment Response to Music Listening Intervention after Stroke"

_brainsci, 2021, doi:10.3390/brainsci11121576_

Round 1
Reviewer 1 Report
See the attached file.

Author Response
Reviewer #1
This interesting article introduce the possibility to find predictive factors of response to the music listening treatment in post-acute stroke patients. The Authors used regression models to correlate baseline demographic, clinical musical and also fMRI data with results concerning verbal memory and language skills.
The article is well structured and well written. It provides an interesting perspective that can be extended to other music therapy interventions and in other clinical settings. The topic of inclusion criteria in music therapy is rather neglected and could benefit from the approach described in this paper.
Some minor remarks:
Comment 1: In general (especially in the Abstract and Discussion) the Authors should reduce a little the magnitude of the results considering the small number of patients included in the sample.
Reply 1: We thank the Reviewer for the positive feedback. We agree that the present analyses are based on limited sample size, which we tried to reflect in the first version of the manuscript. However, we have now further toned down the magnitude of discussion of the results across the revised manuscript. For example, please see:
Abstract, page 1: “Our results suggest that clinical, demographic, and neuroimaging data predicts music listening treatment response. This data could be used clinically to target music-based treatments.”
Discussion, page 11: “The present study extends previous results on the rehabilitative effects of music listening after stroke [13-17] by revealing important information of possible treatment responders. This putative evidence is important in personalizing music-based rehabilitation strategies [1] as well as in improving our understanding of post-stroke language and verbal memory rehabilitation.”
Discussion, page 11: “Our result could be interpreted as indicating that music listening might be particularly beneficial and useful in patients with higher severity of aphasia, but then again it may also reflect the current study population, which did not include patients with global or very severe aphasia. Future studies with larger samples of patients representing the full continuum of aphasia severity are needed to address this issue.”
Discussion, page 12: “Moreover, as the current sample size is limited, future studies with larger samples of patients representing the full continuum of aphasia severity are needed to address this issue.”
Comment 2: Introduction: authors should better contextualize the preferred music listening intervention among possible interventions with music in the stroke setting. This might be helpful in recalling possible active and receptive interventions that can be implemented in stroke rehabilitation.
Reply 2: We have now revised this part of the Introduction accordingly by providing a broader context on the utilization of music in stroke rehabilitation:
Introduction, page 2: “Previously, music has been found to be a useful tool to aid post-stroke cognitive recovery. During the last 20 years, promising evidence has emerged on the efficacy of music-based interventions targeted for specific cognitive, motor, and verbal deficits caused by stroke [1]. These include active interventions that utilize instrument playing or musical rhythm to rehabilitate deficits in motor control of movements (hemiparesis) [13] and singing-based methods to rehabilitate deficits in speech production (aphasia) [14] as well as receptive interventions that utilize music listening.”
Comment 3: Materials and Methods (Intervention): the last sentence is unclear. Between the conclusion of the intervention and follow-up, were the patients free to continue listening to music? If so, could this not constitute a bias from a methodological point of view?
Reply 3: We apologize for the unclear presentation. In the study, the patients listened to the allocated intervention material between the baseline assessment and the 3-month post-stroke stage follow-up (i.e., the intervention period). After this, the patients were given the opportunity to continue listening to the intervention material at their own will. However, the current study is not confounded by this in anyway as the current analyses focus purely on the intervention period (subacute to 3-month stage).
Comment 4: Discussion: The authors should mention other studies in the field of music therapy that, albeit in different ways and in other areas, have aimed to investigate the predictive factors of success of the intervention; the Authors should add a few sentences to underline how this approach could be usefully extended, with the appropriate differences, to other music-therapeutic techniques and other clinical fields.
Reply 4: We thank the reviewer for this suggestion. We would like to note that studies on music-based intervention treatment predictors are extremely scarce. However, we have now added a paragraph in the discussion covering this subject, please see pages 13-14:
“To our best knowledge, treatment response predictors of music-based interventions after a brain injury (e.g., stroke) have not been published nor studies that evaluate neurobiological treatment response predictors. However, the present results might be extended to other forms of music-based interventions such as music therapy where active and receptive music-based interventions are combined and delivered in a clinical setting by a credentialed music therapist. For example, in motor rehabilitation, the initial severity of paresis and activity in the motor and somatosensory cortices during music listening might predict treatment outcome in music therapy or MST. Similarly, in singing‐based speech training interventions, such as melodic intonation therapy (MIT), preserved neural networks subserving language and singing could predict treatment responses. In music therapy, the therapeutic relationship might also play a significant role. For example, in autism spectrum disorder, the therapeutic relationship has been shown to be an important predictor of favourable outcome after music therapy [54]. In contrast, no such influence was identified in individual music therapy for depression, nor did age or musicianship predict significant responses [55]. Evaluating clinical and neurobiological predictors of treatment response to music therapy after stroke would be of great importance for further improving our understanding of stroke rehabilitation with music-based interventions.”
Reviewer 2 Report
The manuscript entitled “Clinical and neural predictors of treatment response to music listening intervention after stroke” is a fair attempt in the field of therapeutic effect of Stroke patients. However, I would like to suggest that the study has minor defects.
- The authors should explain how this study is different from the other studies already conducted.
- A lot of grammatical and typological errors appear in the manuscript. Eg. Line 224, remove “in” in the sentence
- In the method section, the sample used is n=31. Stroke is a very common disorder so can the authors justify why they used only 31 stroke patients?
- The discussion has to be thoroughly checked and has to be amended so as to make it clearer and more specific.
- I recommend the authors should include a graphical abstract to make the understanding easy and clearer.
- In the material and method section, authors have mentioned that they accessed MPI scans at 3 and 6 months post-stroke stage but Figure 1 only presents data for subacute 3 months. Please justify.
On the basis of the above comments
Author Response
Reviewer #2
The manuscript entitled “Clinical and neural predictors of treatment response to music listening intervention after stroke” is a fair attempt in the field of therapeutic effect of Stroke patients. However, I would like to suggest that the study has minor defects.
Comment 1: The authors should explain how this study is different from the other studies already conducted.
Reply 1: We thank the Reviewer for the feedback. To our best knowledge, previous studies on treatment response predictors of music listening intervention after brain injury (e.g., stroke) have not been published nor studies that evaluate neurobiological treatment response predictors in of music-based interventions, receptive or active, in any neurological disease. Few studies have evaluated demographical or treatment-related predictors of music therapy in depression (Erkkilä et al. 2011 British Journal of Psychiatry) and autism spectrum disorder (Mössler et al. 2019 Journal of Autism and Developmental Disorders) but have yielded inconsistent results. We have now included these articles in the Discussion, please see pages 13-14.
If the Reviewer is referring to the previous studies on music listening in the rehabilitation of stroke patients, we would like to clarify the following: While previous studies have shown that daily music listening during the subacute post-stroke stage is beneficial for improving cognitive and emotional recovery (Särkämö et al., 2008 Brain; Baylan et al., 2020 International Journal of Stroke; Sihvonen et al., 2020 Annals of Clinical and Translational Neurology) and in inducing treatment-related structural (Särkämö et al., 2014 Frontiers in Human Neuroscience; Sihvonen et al., 2020 Annals of Clinical and Translational Neurology; Sihvonen et al. 2021 eNeuro) and functional (Sihvonen et al. 2021 eNeuro; Sihvonen et al. 2021 European Journal of Neuroscience) neuroplasticity, these studies have not assessed the neural or behavioural prerequisites of successful music listening based treatment response. This has been stated in the Introduction pages 3-4.
Comment 2: A lot of grammatical and typological errors appear in the manuscript. Eg. Line 224, remove “in” in the sentence
Reply 2: We thank the Reviewer for pointing this out and we have now proofread manuscript.
Comment 3: In the method section, the sample used is n=31. Stroke is a very common disorder so can the authors justify why they used only 31 stroke patients?
Reply 3: We agree with the Reviewer that the sample size is limited, and this limitation is now included in the manuscript (please see page 12). We have also toned down the discussion. However, we would like to note that this study is not an RCT determining whether music listening is effective in supporting cognitive and neural recovery after stroke, rather its aim is to determine if we could predict who benefits from this intervention. We have previously demonstrated that music listening, especially vocal music listening, is an effective tool to support cognitive and neural recovery after stroke as well as to enhance early language recovery in aphasia using data pooled from two single-blind RCTs in stroke patients (N = 83). Moreover, we would like to note that even the largest RCTs to date assessing treatment induced neuroplasticity in aphasia includes 25 to 26 patients (Fridriksson 2010 Journal of Neuroscience; Fleming et al. 2020 Journal of Neurology, Neurosurgery and Psychiatry). Furthermore, studies evaluating the efficacy of singing-based treatments in aphasia rehabilitation, that is, melodic intonation therapy (MIT) include 17 to 30 patients (Halo-Martinez et al. 2021 Frontiers in Neurology). In this light, the current sample size is not that limited. For further information on the patient sample, please see Figure 1: Flow chart outlining the design and progress of the study in Sihvonen et al., 2020 Annals of Clinical and Translational Neurology. In short: The current sample derives from the original 3-arm (vocal music, instrumental music, audiobooks) RCT (ClinicalTrials.gov identifier: NCT01749709) that included 50 patients in total. As music listening has shown to be efficient in improving the cognitive and neural recovery after stroke compared to audiobooks (please see Reply 1), we sought here to identify clinical, demographic, and musical factors and functional activation pattern and network-level factors that predict treatment response of a post-stroke music listening intervention. To do this, we assessed patients in the vocal music and instrumental music groups. Out of these patients, 31 completed the MRI imaging as well as behavioural assessments at both subacute and 3-month follow-up stage and were therefore included in the present analyses for predicting music listening treatment responses.
Comment 4: The discussion has to be thoroughly checked and has to be amended so as to make it clearer and more specific.
Reply 4: We have now self-reviewed the discussion in this light and considered ways of making it clearer and more specific. In our view, the structure of the discussion is quite straightforward and standard. We open with referring to the study aims and pointing clearly out the main findings. Next, we contextualize the current findings in the light of prediction tools for stroke rehabilitation and the lack of research on evaluating treatment response predictors after post-stroke language deficits. The third paragraph discusses the demographic and behavioural findings in appropriate context whereas the fourth and fifth paragraphs discuss the neuroimaging findings. After this, we have added a new paragraph on studies in the field of music-based interventions, including music therapy, investigating the predictive factors of success of the intervention. Due to the scarcity of these studies, we have discussed how the present findings could be extended to other forms of music-based interventions such as music therapy. Lastly, we present conclusions.
If the Reviewer has more specific suggestions on what to add / modify / leave out from the discussion, we are happy continue revising it based on the suggestions.
Comment 5: I recommend the authors should include a graphical abstract to make the understanding easy and clearer.
Reply 5: We thank the Reviewer for this suggestion and have now included a graphical abstract.
Comment 6: In the material and method section, authors have mentioned that they accessed MPI scans at 3 and 6 months post-stroke stage but Figure 1 only presents data for subacute 3 months. Please justify.
Reply 6: We apologize for the unclear presentation. In the study protocol, the patients listened daily (min 1 h/day) to the allocated intervention material between the baseline assessment and the 3-month post-stroke stage follow-up. After the intervention period, the patients were free to continue listening at their own will, without specific requirements on the frequency/intensity of the listening. Therefore, to purely focus on the benefits brought by the music structured listening intervention, we focused on patients who received the music listening treatment and had completed MRI imaging as well as behavioural assessments at both subacute and 3-month follow-up stage. We have now removed the 6-month post-stroke stage from the manuscript, to avoid confusion.